# Spatial Distribution and Estimation Model of Soil pH in Coastal Eastern China

**DOI:** 10.3390/ijerph192416855

**Published:** 2022-12-15

**Authors:** Xiansheng Xie, Jianfei Qiu, Xinxin Feng, Yanlin Hou, Shuojin Wang, Shugang Jia, Shutian Liu, Xianda Hou, Sen Dou

**Affiliations:** 1Guangxi Geographical Indication Crops Research Center of Big Data Mining and Experimental Engineering Technology, Nanning Normal University, Nanning 530001, China; 2Research Institute of Forestry Policy and Information, Chinese Academy of Forestry, Beijing 100091, China; 3Jilin Academy of Agricultural Sciences, Changchun 130033, China; 4School of Geography and Planning, Nanning Normal University, Nanning 530001, China; 5Key Laboratory of Environment Change and Resources Use in Beibu Gulf (Ministry of Education), Nanning Normal University, Nanning 530001, China; 6Guangxi Key Laboratory of Earth Surface Processes and Intelligent Simulation, Nanning Normal University, Nanning 530001, China; 7College of Resource and Environmental Science, Jilin Agricultural University, Changchun 130118, China

**Keywords:** soil health, mean annual precipitation, mean annual temperature, hydrothermal condition, estimation model

## Abstract

Soil pH is an essential indicator for assessing soil quality and soil health. In this study, based on the Chinese farmland soil survey dataset and meteorological dataset, the spatial distribution characteristics of soil pH in coastal eastern China were analyzed using kriging interpolation. The relationships between hydrothermal conditions and soil pH were explored using regression analysis with mean annual precipitation (MAP), mean annual temperature (MAT), the ratio of precipitation to temperature (P/T), and the product of precipitation and temperature (P*T) as the main explanatory variables. Based on this, a model that can rapidly estimate soil pH was established. The results showed that: (a) The spatial heterogeneity of soil pH in coastal eastern China was obvious, with the values gradually decreasing from north to south, ranging from 4.5 to 8.5; (b) soil pH was significantly correlated with all explanatory variables at the 0.01 level. In general, MAP was the main factor affecting soil pH (*r* = −0.7244), followed by P/T (*r* = −0.6007). In the regions with MAP < 800 mm, soil pH was negatively correlated with MAP (*r* = −0.4631) and P/T (*r* = −0.7041), respectively, and positively correlated with MAT (*r* = 0.6093) and P*T (*r* = 0.3951), respectively. In the regions with MAP > 800 mm, soil pH was negatively correlated with MAP (*r* = −0.6651), MAT (*r* = −0.5047), P/T (*r* = −0.3268), and P*T (*r* = −0.5808), respectively. (c) The estimation model of soil pH was: *y* = 23.4572 − 6.3930 × lgMAP + 0.1312 × MAT. It has been verified to have a high accuracy (*r* = 0.7743, *p* < 0.01). The mean error, the mean absolute error, and the root mean square error were 0.0450, 0.5300, and 0.7193, respectively. It provides a new path for rapid estimation of the regional soil pH, which is important for improving the management of agricultural production and slowing down soil degradation.

## 1. Introduction

As an important component of agroecosystems, soils provide a variety of ecosystem services such as carbon stock, water regulation, and food production [1,2]. Soil pH is the indicator to measure soil acidity and alkalinity, which reflects soil fertility and soil health [3]. The change in soil pH affects the chemical solubility and effectiveness of soil nutrients, the activity of soil microorganisms and enzymes, the mobility of heavy metals (such as Cu, Zn, Fe, Mn), and carbon and nitrogen transformation [4,5]. Both soil acidification and salinization are detrimental to proper crop growth. The best growing conditions for most crops are those when the soil pH is 6.5 to 7 [6]. To ensure sustainable agriculture and promote soil health, it is necessary to measure the soil pH regularly. The traditional soil pH measurement methods are mainly based on field sampling, which is usually a labor-intensive task, and the results obtained are considered true values [7]. However, it does not meet the need for accurate management of soil nutrients and guidance of agricultural production, and a more rapid, efficient, and inexpensive method for estimation is required to be invented.

Recently, the digital soil mapping (DSM) method has been widely used to estimate the regional soil pH, which is achieved through partial soil survey data with environmental variables and geostatistical models [8,9]. Remote sensing technology provides a more convenient and efficient tool to obtain the corresponding environmental variables [10]. Of course, some studies have also attempted to directly estimate soil pH by hyper-spectral or visible and near-infrared (VIS/NIR) diffuse reflectance spectroscopy to avoid tedious sampling work [11,12]. Meanwhile, geostatistical models such as kriging, inverse distance weighted (IDW), radial basis function (RBF), and multiple linear regression (MLR), random forests, and geographically weighted regression (GWR) are used to varying degrees [13,14]. With the emergence of more mature methods and new models, some studies have introduced machine learning models for soil pH estimation [15,16]. Some new techniques, such as LIBS (laser-induced breakdown spectroscopy), neural networks, and HSI color image processing are also gradually tried in some studies [17,18,19]. Although the above methods can achieve the comparison and estimation of soil pH changes at the regional scale, there are some limitations, such as relying on soil survey data, more parameters required for the model, and a slow inversion process.

There are many factors influencing soil pH, including climate, soil parent material, and agricultural production activities [2]. Allen et al. [20] believed that at a large scale, the change of soil pH was driven by climatic conditions, especially for topsoil. Furthermore, some studies considered precipitation as the most important environmental factor contributing to spatial variation in soil pH [21,22]. The change of the rainfall patterns on a large scale due to climate change would have a direct impact on soil pH. When precipitation greatly exceeded evaporation, eluviation was very evident. Soil base cations, such as potassium ions (K^+^), calcium ions (Ca^2+^), magnesium ions (Mg^2+^), and sodium ions (Na^+^), tended to move downward or got lost through leaching and were replaced by acid-forming cations, such as hydrogen ions (H^+^), aluminum ions (Al^3+^), and iron ions (Fe^2+^ or Fe^3+^), which led to the evolution of the soil towards acidification [23,24]. Conversely, alkalinization tended to occur. In addition, higher air temperatures could cause higher soil temperatures, which usually increased the chemical reaction rate of solutions, and could enhance soil microbial activity, making the soil buffering capacity decrease and intensifying the leaching of precipitation to the soil [25,26]. In the long term, climate change would also drive changes in soil organic matter status, carbon and nutrient cycling, and effective plant water, which would affect plant productivity and soil pH [20]. However, it is still unclear whether there is a synergistic effect of precipitation and temperature on soil pH, and it also needs to be further verified that precipitation is the dominant environmental factor.

Current climate change is accelerating and the global warming trend is becoming more pronounced. Rainfall over land, including monsoon rainfalls, will become more variable and intense [27]. These will lead significantly to changes in soil processes and properties, adding more uncertainty and complexity to agroecosystems, jeopardizing food security and causing an overall decline in human health [28]. Coastal Eastern China is a major agricultural farming area that plays a key role in maintaining national food security and stability. In 2020, it contributed 40% of national cultivated land area as well as 44% of national grain production [29]. Meanwhile, it presents one of the most complete spectra of climatic zones in the Northern Hemisphere [30]. Under the long-term influence of East Asian monsoon activity, there exist extremely heterogeneous trends in precipitation in time and space and very complex hydrothermal conditions, resulting in a variety of soil types [31,32]. Since the 21st century, climate conditions in eastern China have become more unstable, with greater fluctuation in precipitation [33]. Therefore, it is necessary to analyze the spatial distribution of soil pH in this region and explore a more convenient and economical estimation method to guide the agricultural management. So far, to the best of our knowledge, no studies have considered it as a significant study area and estimated its soil pH. Although some studies provided accurate detailed maps of soil pH for the whole or some regions of China, they were also based on DSM technology and Second National Soil Inventory (1980s) [22,34].

Fortunately, in recent years, China has implemented a Soil Test-Based Formulated Fertilization Project, which has collected a large number of farmland soil samples and provided a good data base for this study. Considering the profound impact of climate change on soil pH, the objectives of this study were: (a) To analyze the spatial distribution characteristics of soil pH and the intrinsic relationships between soil pH and temperature and precipitation at the regional scale based on the farmland soil survey dataset and meteorological dataset; (b) to estimate soil pH directly from precipitation and temperature by linear regression model, and provide a potential tool that does not depend on a large number of soil survey points and can rapidly estimate regional soil pH with few input parameters, reducing the data collection burden and avoiding uncertainty in the DSM process. We believed that this method could provide a proper soil quality assessment for coastal Eastern China, which would be important for improving the management of agricultural production, slowing down soil degradation and maintaining food security.

## 2. Materials and Methods

### 2.1. Study Area

Coastal eastern China referred to in this study includes 13 provincial districts (18.16° N–53.56° N, 108.62° E–135.08° E), from north to south: Heilongjiang, Jilin, Liaoning, Hebei, Beijing, Tianjin, Shandong, Jiangsu, Shanghai, Zhejiang, Fujian, Guangdong, and Hainan, covering a total land area of 172 × 10^4^ km^2^ (Figure 1). It covers five climatic zones: cold temperate, middle temperate, warm temperate, subtropical, and tropical, spatially spanning about 5000 km, with soil types gradually transitioning from black soil and cinnamon soil to yellow soil and red soil. The region is located in the Third Step of China, abundant in broad plains, and dotted with foothills and lower mountains, with altitudes of below 1000 m. The monsoon climate is significant. In summer, the temperature is high and the precipitation is high, too, mainly from May to September. In winter, most areas are cold and dry due to the influence of the Mongolian/Siberian high-pressure system. The mean annual temperature (MAT) ranges from 3.69 °C to 24.57 °C and the mean annual precipitation (MAP) ranges from 507.55 mm to 1765.88 mm, showing an increasing trend from north to south (Table 1). The major land cover types are croplands, forestland, grassland, and construction land. By 2020, the population of the region is 663 million, which represents 47% of the national population [29].

### 2.2. Data Sources

In 2005, China started to implement the Project of Soil Test-Based Formulated Fertilization which included soil physical and chemical property testing as one of the basic tasks. During the years 2000 to 2014, a large number of soil samples were collected from grain fields, vegetable fields, and orchards, and tested for indicators such as soil organic matter, total nitrogen, effective phosphorus, fast-acting potassium, and soil pH. The samples were taken within the 0–20 cm depth (topsoil). Soil pH was determined by potentiometric method (water extraction, 1:2.5 ratio). Although the timing of the soil tests varied from place to place, most of the sampling was completed around 2008. In 2014, after organizing experts to review the soil basic test data and spatial data, the National Agricultural Technology Extension Service Center edited and published the Soil Basic Nutrient Data Set (2005–2014), which reflected the average level of soil status of each county for the period. In this study, we selected soil pH data belonging to coastal Eastern China from the dataset and obtained a total of 778 sample points. The GPS coordinates of the county administrative centers were used as the geographic coordinates of the sample points (Figure 1).

The meteorological dataset was obtained from the National Meteorological Science Data Center (http://data.cma.cn (accessed on 10 June 2022)), with MAP (mm) and MAT (°C) as the main indicators. To reduce the interference of the results in the years of climate anomalies, we extended the time series of meteorological data appropriately. Considering the completion time of sampling, meteorological data from 2000 to 2016 for each county in coastal Eastern China were selected and the average values of this period were matched with the corresponding soil pH data.

### 2.3. Data Processing and Analysis

Considering the original data of MAP was large, it was logarithmically processed (to the base 10) to achieve better stationarity. Referring to Sun and Du [35], we introduced the aridity index (the ratio of precipitation to temperature, P/T) as a new variable. At the same time, to analyze the synergistic effect of precipitation and temperature on soil pH, we defined a new variable, the product of precipitation and temperature (P*T). And the same logarithmic treatment was applied to P/T and P*T, which were calculated as follows:(1)P/T=lg(MAP/MAT)
(2)P∗T=lg(MAP×MAT)

According to the Technical Specification for Soil Test-Based Formulated Fertilization and the characteristics of the study area, the acidity and alkalinity of the soil were classified as highly acidic grade (4.5 < pH ≤ 5), acidic grade (5 < pH ≤ 5.5), weakly acidic grade (5.5 < pH ≤ 6.5), neutral grade (6.5 < pH ≤ 7.5), and weakly alkaline grade (7.5 < pH ≤ 8.5). To display the spatial distribution characteristics of pH in coastal Eastern China during this period visually, we used the ordinary kriging method for spatial interpolation, because it is the best linear unbiased estimator based on the semi-variogram model [36], and the specific principle and method can be found in the work of Arslan, et al. [37]. This study was implemented through the geostatistical tools of ArcGIS software (version 10.5). The semi-variogram model is as follows:(3)$$\gamma_{h}=\frac{1}{2n}\overset{n}{\underset{i=1}{\sum }}{\left[Z\left(x_{i}\right)-Z\left(x_{i}+h\right)\right]}^{2}$$
where, $\gamma_{h}$ is the semi-variance value for all pairs at a lag distance  h; n represents the number of pairs of observations separated by the distance  h, $Z\left(x_{i}\right)$ is the measured value of the variable at point $x_{i}$, $Z\left(x_{i}+h\right)$ is the measured value of the variable at point $x_{i}+h$.

The 800 mm precipitation contour is considered to be an important geographic boundary between southern and northern China, as well as the boundary between humid and semi-humid regions and between subtropical and temperate monsoon climate zones [38,39]. Besides studying the general relationship between hydrothermal conditions and soil pH, we also divided the study area into regions with MAP < 800 mm and regions with MAP > 800 mm to explore the differences in relationship between hydrothermal conditions and soil pH in different regions. Notably, under the combined influence of the East Asian Monsoon and topography, some areas east of the Changbai Mountain are located on the windward slope and received more topographic rainfall, with precipitation above 800 mm, which is considerably higher than that near the same latitude [40,41]. These areas mainly include Ji’an City, Linjiang City, Helong City, Fusong County, Jinyuan County, Tonghua County in Jilin Province, Zhenxing District, Zhen’an District, Donggang City, Fengcheng City, Huanren County, and Kuandian County in Liaoning Province. To analyze the effect of the mentioned areas on the statistical results, the differences before and after removal were compared.

Building and validation of pH estimation model: principal component analysis (PCA) is a method to reduce the number of variables [42]. More details on PCA can be found in the work of Zitko [43] and Jolliffe and Cadima [44], which provides a good overview. Based on the PCA results, the variables with eigenvalues greater than 1 were identified as principal components and used for modeling. 70% of the sample points were randomly selected to determine the model parameters by multiple linear regression and the least square method, and the remaining 30% of sample points (those not involved in the modeling) were used to verify the accuracy of the model. Mean error (ME), mean absolute error (MAE), and root mean square error (RMSE) were selected to assess the difference between the estimated and true values. These formulas are as follows:(4)$$\mathrm{ME}=\frac{1}{N}\overset{N}{\underset{i=1}{\sum }}\left[S\left(x_{i}\right)-\left(T_{i}\right)\right]$$
(5)$$\mathrm{MAE}=\frac{1}{N}\overset{N}{\underset{i=1}{\sum }}\left|S\left(x_{i}\right)-\left(T_{i}\right)\right|$$
(6)$$\mathrm{RMSE}=\sqrt{\frac{1}{N}\overset{N}{\underset{i=1}{\sum }}\left[\left(x_{i}\right)-{\left(T_{i}\right)}^{2}\right]}$$
where N is the number of verification points, $S\left(x_{i}\right)$ is the true value of the variable at point $x_{i}$, and $T\left(x_{i}\right)$ is the estimated value of the variable at point $x_{i}$.

The Pearson correlation coefficient (*r*) was used to measure the correlation between different variables, and statistical tests were performed using IBM SPSSStatistics software (version 22). The plotting and data processing were performed using Sigmapolt software (version 14.0) and Microsoft Excel software (version 2010).

## 3. Results

### 3.1. Spatial Distribution of Soil pH in Coastal Eastern China

There was an obvious spatial heterogeneity in soil acidity and alkalinity in coastal Eastern China, with alkaline in the north and acidic in the south. In general, the soil pH value gradually increased from south to north, ranging from 4.5 to 8.5 (Figure 2).

Northeast China is located in the cold-temperate and mid-temperate climate zones. In summer, it is warm and rainy; in winter, it is cold and dry. Under the long-term influence of this climatic condition, the typical soils of black soil and Chernozem have been formed, and a small amount of albic soil is also found. The black soil is clayey with a good water-retentive ability and poor aeration, showing a neutral or weakly acidic property. Chernozem is easy to form a calcic horizon under the influence of eluviation-illuviation, which can obtain more basic cations and show a neutral or weakly alkaline property. The basic cations of albic soil are easy to get lost with periodically waterlogged conditions, showing an acidic property. Therefore, the acidity and alkalinity of soils in Heilongjiang, Jilin, and Liaoning were mainly in the weakly acidic, neutral, and weakly alkaline grades. The soil pH in Heilongjiang ranged from 5.43 to 7.85 with an average of 6.19, showing an increasing trend from northeast to southwest. The soil pH in Jilin ranged from 5.48 to 8.50 with an average of 6.54, showing an increasing trend from southwest to northeast. The soil pH in Liaoning ranged from 5.60 to 7.56 with an average of 6.41, showing an increasing trend from east to west.

Hebei, Beijing, and Tianjin belong to North China and are located in the warm temperate climate zone. There is little precipitation throughout the year, and the seasonal distribution is uneven. Under this condition, cinnamon soil, calcareous fluvo-aquic soil, and coastal saline soil are formed. Cinnamon soil is moderate in texture, less affected by eluviation, and accumulate a certain amount of calcium carbonate, showing a neutral or weakly alkaline property. Calcareous fluvo-aquic soil is affected by slight eluviation-illuviation, and coastal saline soil has a large salt content, both of which show a neutral or weakly alkaline property. Therefore, the acidity and alkalinity of soils in these areas were mainly in the neutral and weakly alkaline grades. The soil pH in Hebei ranged from 6.89 to 8.41 with an average of 7.82, showing a northeast-to-southwest increasing trend. The soil pH in Beijing ranged from 7.10 to 8.31 with an average of 7.68, showing a northeast-to-southwest increasing trend. The soil pH in Tianjin ranged from 7.51 to 8.50, with an average of 8.19, and was alkaline throughout the region.

Shandong, Jiangsu, Shanghai, Zhejiang, and Fujian belong to East China, where the climate transitions from a warm temperate monsoon climate to a subtropical monsoon climate, resulting in a rich diversity of soil types. (a) The soil pH in Shandong ranged from 5.01 to 8.41, with an average of 7.05, showing an increasing trend from east to west. Its eastern part was mainly brown soil and chisley soil. Both of the soils were constantly subjected to the action of infiltrating water during soil formation leading to an increase in the content of exchangeable acidity, and the soil was in the acidic or weakly acidic grade. Its central and western parts were mainly cinnamon soil, in a neutral or weakly alkaline grade. (b) Jiangsu was located in the middle and lower reaches of the Yangtze River, with the soil pH ranging from 5.99 to 8.07 and an average of 7.16, showing an increasing trend from south to north. Paddy soil was widespread, evolving toward neutrality through eluviation and illuviation. Its central and eastern parts were mainly paddy soil and cinnamon soil, in neutral and weakly alkaline grades. Its southern part was mainly yellow-brown earth. Affected by the aluminizing process, the soil was weakly acidic. (c) The soil type and distribution characteristics of Shanghai were similar to those of Jiangsu, with the soil pH ranging from 5.97 to 8.19 and an average of 7.23. (d) The soil pH in Zhejiang ranged from 5.03 to 6.40, with an average of 5.52, showing an increasing trend from south to north. Its northern part was mainly weakly acidic paddy soils. Its southern part was dominated by red soil and yellow soil, which were acidic under the strong leaching effect. (e) The soil pH in Fujian ranged from 4.87 to 5.42, with an average of 5.07, showing a decreasing trend from east to west. It was also dominated by red soil and yellow soil. Its western part is Wuyi Mountain, which is on the windward slope of the East Asian monsoon with more precipitation. The soil of this area was subjected to strong leaching and showed a highly acidic property.

Guangdong and Hainan are part of Southern China, serving as a transition zone from the subtropical monsoon climate to the tropical monsoon climate. It is characterized by warm and rainy, abundant light and heat, and long summers. With this climate, the soils of Guangdong were mainly red soil and lateritic red soil, mostly in the acidic grade, and the rest in the weakly acidic grade, with pH ranging from 5.14 to 5.72 and an average of 5.41. The weakly acidic soils were mainly located in some economically developed areas, such as the Pearl River Delta region and Chaoshan region, which were strongly influenced by human activities, such as a series of agricultural management measures to improve the soil. The soils of Hainan were mainly Latosol and lateritic red soil, mostly in the acidic and highly acidic grade, and the rest in the weakly acidic grade, with pH values ranging from 4.70 to 5.70 and an average of 5.13. Its central part is Wuzhishan Mountain, which is also on the windward slope with more precipitation. The soil of this area was subjected to strong leaching and showed a highly acidic property. The soil of the Changjiang area was weakly acidic, possibly because its topography was mainly plain and located at the estuary of the river, which could take in more saline material flowing in from higher places and had higher soil pH.

### 3.2. Effects of Hydrothermal Conditions on Soil pH

#### 3.2.1. The General Relationship between Soil pH and Hydrothermal Conditions

In general, soil pH was significantly negatively correlated with MAP, MAT, P/T, and P*T at the 0.01 level (Figure 3). Soil pH had the highest correlation with MAP (*r* = −0.7244), and the expression for the relationship was: *y* = −3.8031*x* + 17.731. The correlation coefficient between soil pH and MAT was −0.3367, and the expression of the relationship was: *y* = −0.0681*x* + 7.6062. The correlation coefficient between soil pH and P/T was −0.6007, and the expression of the relationship was: *y* = −3.418*x* + 12.916. The correlation coefficient between soil pH and P*T was −0.4654, and the expression of the relationship was: *y* = −1.271*x* + 11.747. It showed that the aridity of the climate significantly affected the soil pH: the drier the climate, the greater the pH value and the evolution of the soil in the alkaline direction. It could also be found that temperature also had an effect on soil pH, and precipitation and temperature showed synergistic effects to some extent, but still precipitation was the dominant factor.

#### 3.2.2. Relationship between Soil pH and Hydrothermal Conditions in Regions with Different MAP

As shown in Table 2, in the regions with MAP < 800 mm, soil pH showed a significant negative correlation with MAP (*r* = −0.4631), but a significant positive correlation with MAT (*r* = 0.6093). It indicated that temperature had a greater effect on soil pH than precipitation in these regions. As the temperature increased, the soil pH value became larger. This was mainly because in the areas where precipitation was commonly low, rising temperatures would increase the water deficit in the soil, which led to the soil becoming drier, the soil buffering capacity increasing, base cations being able to be fixed, and the soil evolving in an alkaline direction. Soil pH was significantly negatively correlated with P/T (*r* = −0.7041) but positively correlated with P*T (*r* = 0.3951). It indicated that the aridity of the climate still significantly affected the soil pH, obeying the general relationship described above.

In the regions with MAP > 800 mm, soil pH was significantly negatively correlated with MAP, MAT, P/T, and P*T, which were consistent with the general relationship, with correlation coefficients of −0.6651, −0.5047, −0.3268, and −0.5808, respectively. It showed that under the condition of high precipitation in general, eluviation played a major role in impacting soil pH. Precipitation could not only produce a strong driving force for the migration of soil base cations but also acted as a “pipe” role, making them easy to migrate downward with the water or be lost. As the soil salt base saturation decreased, the soil evolved toward acidification. With the decrease of the base saturation level on the soil exchange sites, the soil evolved toward acidification. At the same time, the high temperature would increase soil microbial activity, which made soil buffering capacity decrease and enhanced leaching. After removing the 12 outliers, the correlations between pH and MAP, MAT, P/T, and P*T were found to be enhanced with correlation coefficients of −0.7029, −0.6262, −0.3912, and −0.7156, respectively. The correlation between soil pH and P*T was significantly improved, which better reflected the important influence of the synergistic effect of hydrothermal conditions on soil pH in these regions.

### 3.3. The Estimation Model of Soil pH

The results showed that the top two factors with eigenvalues greater than 1 were MAP and MAT (Table 3). The MAP explained 68.87% of the total variation and the MAP explained 30.13%. The cumulative contribution rate of the two reached 99.00%. Therefore, MAP and MAT were finally taken as the principal components of the estimation model. The initial expression of the model can be defined as:(7)y=α+β·lgMAP+γ·MAT
where α, β, and γ are constants. lg indicates taking logarithm (with base 10).

The results of the multiple regression analysis showed that the multiple correlation coefficient (*r*) was 0.82, the goodness-of-fit (*R*^2^) was 0.67, and the standard error was 0.65. As the ANOVA results are shown in Table 4, the F-statistic value was 563.52 (*p* < 0.01). It showed that the model built was credible. The result of the regression statistics was shown in Table 5. According to it, we obtained the expressions of the model as follow:(8)y=23.4572−6.3930×lgMAP+0.1312×MAT

Since the variables of the model were the average of meteorological data over a certain period and the estimates obtained were the average results of soil pH in a certain region under this condition, we redefined it as the regional soil pH constant (*K*) as follow:(9)K=y=23.4572−6.3930×lgMAP+0.1312×MAT

The model was tested to show a good estimation performance (Figure 4). The correlation coefficient was 0.7743 (*p* < 0.01). The ME, MAE, and RMSE were 0.0450, 0.5300, and 0.7193, respectively. It showed that the model had good unbiasedness and the estimated values were pretty close to the true values.

## 4. Discussion

Many studies believed that soils in the eastern part of China showed the spatial characteristics of “acidic in the south and alkaline in the north” [22,45], which was confirmed by the results of this study. It was formed after a long period (hundreds or even thousands of years) of the macro-geological cycle and biological cycle acting under the conditions of a large difference in climate between the north and the south [46,47]. At the regional scale, soil pH tended to be a soil property that developed under natural conditions and had a certain stability [48]. This study also confirmed that MAP was the main environmental variable for estimating soil pH and found that the synergistic effect of precipitation and temperature differed over regions with different MAP. Highly acidic soils (e.g., red soil) were formed in Southern China under long-term hot and rainy conditions. In northern China, under the long-term cold and dry conditions, alkaline cations were easily enriched on the soil surface to form neutral or alkaline soils (e.g., cinnamon soil). Besides climatic factors, soil texture and bulk density also had important effects on soil pH [49]. On the one hand, soil moisture was a key bridge in regulating soil base-forming and acid-forming cations [50]. On the other hand, the response of the soil to additional inputs of acidic material depended on the soil buffering capacity [51]. Both of them were closely related to soil texture and bulk density, and soils with high amounts of clay or organic matter had a higher cation exchange capacity and buffering capacity than sandy soils. In addition, topography could induce spatial and temporal variability in soil chemistry by changing the redistribution of parent material and hydrothermal conditions [52,53]. Soil pH was also affected by other natural factors, such as soil parent material, soil type, and vegetation cover [2,54].

The essence of the change in soil pH was that the soil acid-base balance was severely disturbed. Guo et al. [55] reported that the soil pH in major crop production regions in China showed a significant decreasing trend from the 1980s to the beginning of the 21st century, especially in paddy soils. 90% of the farmlands in China suffered from soil acidification, and it was particularly prominent in the south, where most soil pH values were already below 5.5 and the area and intensity of acidification were still further increasing [56]. In general, the acidity and alkalinity of the topsoil of the farmland would first inherit the characteristics of the parent soil [57]. It was likely to be closely tied to human activities when it changed significantly in a short period. Many studies indicated that the increase in acid rain and excessive application of nitrogen fertilizers were important factors leading to soil acidification [58,59]. Acid rain itself contained a large amount of H^+^, and the ammonium ions in nitrogen fertilizers also released large amounts of H^+^ through nitrification reactions. These H^+^ were adsorbed on the soil surface by exchanging reactions with base cations, while the base cations were leached out with water, eventually leading to soil acidification. Since the reform and opening up, eastern China has experienced continued rapid economic development, accelerated urbanization, industrialization, and population growth, which, on the one hand, led to an increasing demand for energy consumption and large quantities of fossil fuels, which would exacerbate the frequency of acid rain [60], and on the other hand, led to the increasing demand for food and the excessively using nitrogen fertilizer to increase crop yield [61,62]. After 2010, the pH value of Chinese farmland soils showed an increasing trend, and the proportion of acidic soils decreased in the last decade, which was closely related to the full implementation of soil testing and formulated fertilization technology, the vigorous promotion of returning straw to the fields and the zero-growth program of chemical fertilizer use [63]. Soil testing and formulated fertilization technology provided an excellent fertilizer proportioning scheme, which could improve the fertilizer utilization rate and achieve the effect of chemical fertilizer reduction. Straw was rich in nutrients such as nitrogen, phosphorus, and potassium. Returning straw to the fields could effectively promote crop root development, enhance soil buffering performance, and compensate for the loss of exchangeable base cations. To achieve the target of zero growth of chemical fertilizer use, the government organized skills training to raise farmers’ awareness and ability in scientific fertilizer application and launched an action to replace chemical fertilizers with organic fertilizers. In addition, land use changes could alter the structure and function of the previous ecosystem and affect the processes of transport and re-distribution of soil properties [34,64]. Even under the same land use system, unreasonable farming systems and crop management could also lead to inefficient fertilizer use which causes soil acidification [49,58]. Agricultural and economic policies could enhance the modification of soil pH by influencing land use changes. For example, some supportive agricultural policies stimulated farmers to produce, which expanded arable land area, promoted intensive management, and adjusted soil properties [65]. Since the 1980s, the Chinese government introduced a system of household contract responsibility, in which remuneration was linked to output. Intensive cultivation became the major farming method. The grain price protection policy could enhance farmers’ confidence in production by setting a minimum grain purchase price. Through financial support, the agricultural subsidies policy could help farmers to expand production and offset the cost of soil improvement (e.g., lime application).

In summary, the differences and variations in soil pH were the results of a combination of natural and human factors. Natural factors provided the potential environment for soil pH, while human factors could accelerate the change process. There were still some limitations in the soil pH estimation model based on hydrothermal conditions in this study, and there were some errors between the estimated values and the true values. After all, the climate is not the only factor that affects soil pH. The model still needs to be revised in future studies by taking into account the soil texture, topography, and human factors (such as agricultural production activities) to enhance the interpretability. The results obtained in this study were based on currently available data, limited to this study area, and should be applied with caution when generalizing. Certainly, for other regions with similar hydrothermal conditions, this method can be referred to and applied in verifying its rationality. We still believed that this study provided an important idea for soil pH estimation, which attempted to understand the general trend of climate variables affecting soil pH in regional scale for guidance of agricultural production. In the specific application of the model, more realistic local “details”, such as the above-mentioned policy, and economic and land use change factors, can be added to adjust the model parameters and factor weights for improving the estimation accuracy at the microscopic scale. In addition, long-term soil location monitoring should also be strengthened, and the model should be continuously optimized using historical data, thus improving its accuracy and applicability. In the context of accelerated climate change, this study also enlightens that people need to pay more attention to the analysis of meteorological data and formulate agricultural production measures based on local climatic conditions, which ultimately serve to effectively regulate soil pH in farmland.

## 5. Conclusions

In this study, the spatial distribution characteristics of soil pH and the relationship between hydrothermal conditions and soil pH in coastal Eastern China were analyzed based on the Chinese farmland soil survey dataset and meteorological dataset, with MAP, MAT, P/T, and P*T as the main explanatory variables. Based on this, a model that could rapidly estimate soil pH was established. The main conclusions were as follows: (a) The spatial heterogeneity of soil pH in coastal Eastern China was obvious, with alkaline in the north and acidic in the south. (b) In general, MAP was the dominant environmental factor affecting soil pH. The effects of temperature and precipitation on soil pH were interactive rather than independent of each other and differed over different MAP regions. In the regions with MAP < 800 mm, soil pH was negatively correlated with MAP and P/T, respectively, and positively correlated with MAT and P*T, respectively. In the regions with MAP > 800 mm, soil pH was negatively correlated with MAP, MAT, P/T, and P*T, respectively. (c) The estimation model of soil pH was: y = 23.4572 − 6.3930 × lgMAP + 0.1312 × MAT. It was verified to have high accuracy but still needed to be corrected by considering other factors.

It is worth looking forward to the ongoing Third National Soil Survey in China, which will make it possible to obtain more, updated, and accurate soil survey data for model validation and optimization in the future.

## Figures and Tables

**Figure 1 ijerph-19-16855-f001:**
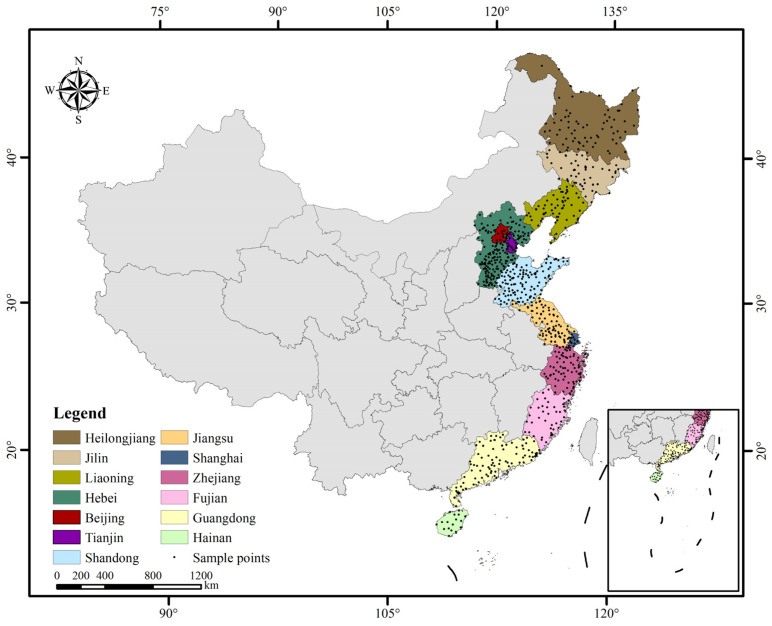
Study areas and sample points distribution.

**Figure 2 ijerph-19-16855-f002:**
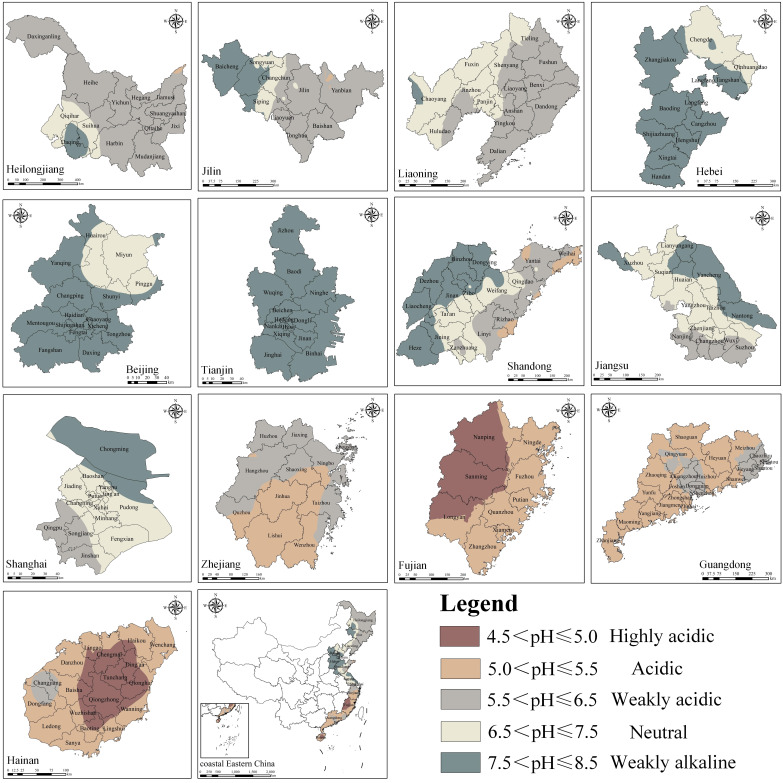
Spatial distribution of soil pH for each provincial district.

**Figure 3 ijerph-19-16855-f003:**
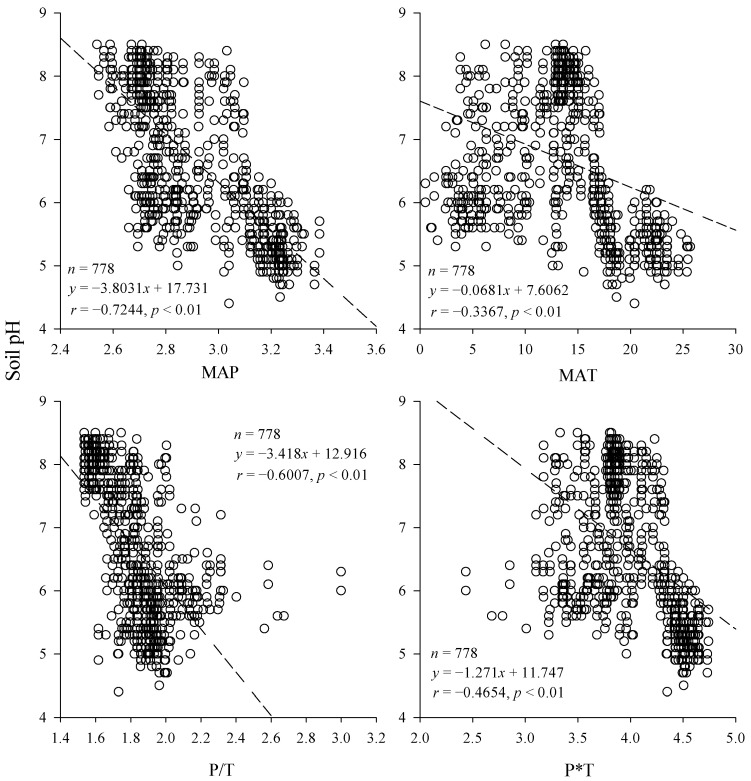
Relationship between soil pH and explanatory variables.

**Figure 4 ijerph-19-16855-f004:**
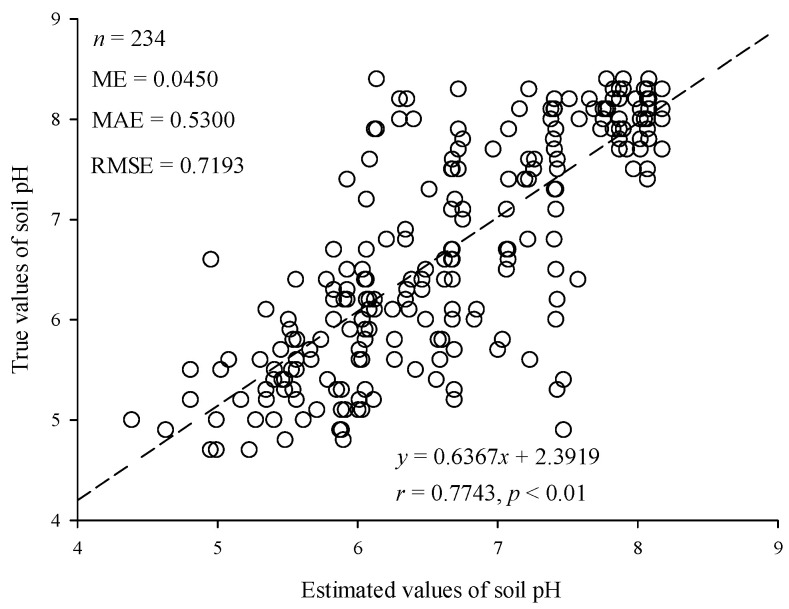
The estimated and true values of the model.

**Table 1 ijerph-19-16855-t001:** Hydrothermal conditions in different provinces.

Provinces	Number of Samples	Mean Annual Precipitation/mm				Mean Annual Temperature/°C			
		Max	Min	Mean	SD	Max	Min	Mean	SD
Heilongjiang	67	627.81	384.41	524.44	57.50	5.30	0.52	3.69	1.19
Jilin	55	922.15	345.87	592.14	148.30	7.65	2.99	5.68	0.88
Liaoning	61	1074.22	456.65	655.87	139.88	11.40	6.29	8.96	1.25
Beijing	9	619.17	528.04	588.79	45.57	13.13	11.43	12.00	0.85
Tianjin	10	539.95	512.27	517.81	11.67	13.27	12.90	12.97	0.16
Hebei	175	665.13	367.55	507.55	60.57	14.59	3.85	12.46	2.30
Shandong	100	1042.03	499.60	675.77	111.53	14.82	6.03	13.43	1.05
Shanghai	9	1238.71	1156.11	1202.00	43.54	17.09	17.02	17.06	0.04
Jiangsu	75	1250.87	845.09	1056.09	132.59	17.02	14.13	15.68	0.85
Zhejiang	74	1722.80	1276.17	1485.87	133.43	18.81	16.60	17.73	0.65
Fujian	41	1861.69	1098.37	1525.52	236.35	21.95	12.61	19.40	2.02
Guangdong	85	2425.68	1233.35	1703.81	243.43	23.50	19.92	22.02	1.02
Hainan	17	2320.43	1050.52	1765.88	409.27	25.69	23.27	24.57	0.83

**Table 2 ijerph-19-16855-t002:** Relationships between soil pH and hydrothermal conditions in different MAP areas.

Different MAP Regions	<800 mm	>800 mm	>800 mm (Outliers Removed)
	Heilongjiang, Liaoning, Jilin, Hebei, Beijing, Tianjin, Some Regions of Shandong	Some Regions of Heilongjiang, Some Regions of Liaoning, Some Regions of Shandong, Zhejiang, Fujian, Shanghai, Jiangsu, Guangdong, Hainan	Some Regions of Shandong, Zhejiang, Fujian, Shanghai, Jiangsu, Guangdong, Hainan
Number of samples	448	330	318
Relationship with MAP	*y* = −5.802*x* + 23.143 *r* =−0.4631 **	*y* = −5.2804*x* + 22.444 *r* = −0.6651 **	*y* = −5.8672*x* + 24.324 *r* = −0.7029 **
Relationship with MAT	*y* = 0.1424*x* + 5.800 *r* = 0.6093 **	*y* = −0.1183*x* + 8.0673 *r* = −0.5047 **	*y* = −0.1798*x* + 9.2693 *r* = −0.6262 **
Relationship with P/T	*y* = −2.6923*x* + 12.047 *r* = −0.7041 **	*y* = −3.2862*x* + 12.091 *r* = −0.3268 **	*y* = −4.7658*x* + 14.836 *r* = −0.3912 **
Relationship with P*T	*y* = 1.4168*x* + 2.0132 *r =* 0.3951 **	*y* = −2.4984*x* + 16.851 *r* = −0.5808 **	*y* = −3.7572*x* + 22.455 *r* = −0.7156 **

** Significant at 0.01 level.

**Table 3 ijerph-19-16855-t003:** Eigenvalues and contributions of principal components.

Factor	Eigenvalue	Contribution Rate/%	Cumulative Contribution Rate/%
MAP	2.76	68.87	68.87
MAT	1.21	30.13	99.00
P/T	0.04	1.00	100.00
P*T	0.00	0.00	100.00

**Table 4 ijerph-19-16855-t004:** The ANOVA results for the model.

Source	df	SS	MS	F	*p*-Value
Between samples	2.00	469.81	234.90	563.52	0.00
Within samples	541.00	225.52	0.42		
Total	543.00	695.32			

**Table 5 ijerph-19-16855-t005:** The regression statistics results for the model.

Variables	Coefficients	Standard Error	t-Stat	*p*-Value	Lower 95%	Upper 95%
Intercept	23.4572	0.5175	45.3269	0.0000	22.4406	24.4738
MAP	−6.3930	0.2056	−31.0990	0.0000	−6.7968	−5.9892
MAT	0.1312	0.0080	16.3692	0.0000	0.1154	0.1469

## Data Availability

The meteorological dataset is available with permission of the China Meterological Administration, which can be found here: http://data.cma.cn (accessed on 10 June 2022). The Soil Data Set is available with permission of the China Agriculture Press, which can be found from a book called The Soil Basic Nutrient Data Set (2005–2014).

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
