# Peer review of "Spatial Distribution and Estimation Model of Soil pH in Coastal Eastern China"

_ijerph, 2022, doi:10.3390/ijerph192416855_

Round 1

Reviewer 1 Report

Soil pH, an important trait for soil property, can be obtained conventionally by field sampling and testing, but such work is time-consuming and lack of continuous data temprally and spatially. This study tried to provide an effective tool in modeling and estimating soil pH by MAP and MTP, which indeed correlated with soil pH.  However, soil pH is determined by several variables, such as parental materials, temperature, precipitation, land use, and we can not quantify the weight or contribution of each parameter for soil pH. Therefore, I'm not sure about the certainty of the model in other areas, which could be evaluated in the current study.  Moreover, numerous syntax errors existed in the present draft. I recommend the authors to make a thorough revision on the English written and recalibrate the logical flow for the whole text. The format of references was not consistent.

Reviewer 2 Report

In this study, the spatial distribution characteristics of soil pH in coastal Eastern China were analyzed using kriging interpolation, and the relationships between hydrothermal conditions and soil pH were explored using regression analysis. The method is appropriate. The study has some significance in content. Some suggestions below:

The word of the title was misspelling(Ph).

The geographical conditions of the study area should be described in detail.

The authors should give a spatial distribution of soil pH for the whole Study area

Reviewer 3 Report

1. The introduction section should clarify current research gaps and summarize its research motivation clearly.

2. Only hydrothermal conditions were examined, and however humans can do little to change these conditions. Thus, I think the authors should focus more on policy factors, economic factors and human behavior factors.

3.The data was collected around 2008. It would be better if data in recent years can be used.

4. The authors should provide more descriptive statistics or pattern analysis of hydrothermal conditions in different regions.

5. The English language should be edited by a native speaker.

Round 2

Reviewer 3 Report

It would be better if the authors can discuss the policy implications of their empirical findings in detail.

Author Response

Thank you for your suggestions. We have added relevant content to the second paragraph of the discussion section to make the policy implications more specific. Please see the red font in the revised manuscript for details.
